# Mi-CARE: Comparing Three Evidence-Based Interventions to Promote Colorectal Cancer Screening among Ethnic Minorities within Three Different Clinical Contexts

**DOI:** 10.3390/ijerph20227049

**Published:** 2023-11-10

**Authors:** Karriem S. Watson, Katherine Y. Tossas, Yazmin San Miguel, Nicole Gastala, Liliana G. San Miguel, Scott Grumeretz, Vida Henderson, Robert Winn, Masahito Jimbo, Keith B. Naylor, Megan E. Gregory, Yamilé Molina, Ashley M. Hughes

**Affiliations:** 1National Institutes of Health, 9000 Rockville Pike, Bethesda, MD 20892, USA; karriem.watson@nih.gov; 2VCU Massey Comprehensive Cancer Center, 417 N 11th St., Richmond, VA 23219, USA; katherine.tossas@vcuhealth.org (K.Y.T.); robert.winn@vcuhealth.org (R.W.); 3Abbott Laboratories, 100 Abbott Park Road, Abbott Park, IL 60064, USA; ysanmiguel@gmail.com; 4Hospital & Health Sciences Systems Mile Square Health Center, University of Illinois, 1220 S Wood St. M/C 698; Chicago, IL 60612, USA; reizinee@uic.edu; 5Department of Community Health Sciences, School of Public Health, University of Illinois Chicago, 1603 W Taylor Street, Chicago, IL 60612, USA; lsanmig7@gmail.com (L.G.S.M.); ymolin2@uic.edu (Y.M.); 6Cancer Center, University of Illinois, SRH MC 709, 818 South Wolcott Avenue, Chicago, IL 60612, USA; sgrumeretz@gmail.com; 7Fred Hutchinson Cancer Center, 1100 Fairview Ave. N. M3-B232, Seattle, WA 98109, USA; vahender@fredhutch.org; 8Department of Family and Community Medicine, College of Medicine, University of Illinois Chicago, 1919 W. Taylor Street, MC 663, Chicago, IL 60612, USA; mjimbo@uic.edu; 9Department of Clinical Medicine, Division of Gastroenterology and Hepatology, College of Medicine, University of Illinois Chicago, 840 S. Wood St., 718E CSB (MC 716), Chicago, IL 60612, USA; knaylor@uic.edu; 10Department of Health Outcomes and Biomedical Informatics, University of Florida College of Medicine, 2004 Mowry Road, Gainesville, FL 32611, USA; megan.gregory@ufl.edu; 11Department of Biomedical and Health Information Sciences, School of Applied Health Sciences, University of Illinois Chicago, 1919 W. Taylor Street MC 530, Chicago, IL 60612, USA; 12Center for Innovation for Chronic, Complex Healthcare, Edward Hines JR VA Hospital, 5000 South 5th Avenue, Bldg 1, Hines, IL 60141, USA

**Keywords:** colorectal cancer, screening disparities, lay navigator, education

## Abstract

Multiple evidence-based interventions (EBIs) have been developed to improve the completion of colorectal cancer (CRC) screening within Federally Qualified Health Centers (FQHCs) and other safety net settings in marginalized communities. Little effort has been made, however, to evaluate their relative effectiveness across different clinical contexts and populations. To this end, we tested the relative effectiveness of three EBIs (mailed birthday cards, lay navigation, and provider-delivered education) among a convenience sample of 1252 patients (aged 50–75 years old, who were due for CRC screening and scheduled for a visit at one of three clinics within a network of Federally Qualified Health Centers (FQHCs) in the United States. To be eligible for the study, patients had to identify as African American (AA) or Latino American (LA). We compared the effects of the three EBIs on CRC screening completion using logistic regression. Overall, 20% of the study population, an increase from a baseline of 13%, completed CRC screening. Clinical demographics appeared to influence the effectiveness of the EBIs. Mailed birthday reminders appeared to be the most effective within the multi-ethnic clinic (*p* = 0.03), provider-delivered education within the predominantly LA clinic (*p* = 0.02), and lay navigation within the predominantly AA clinic (*p* = 0.03). These findings highlight the importance of understanding clinical context when selecting which evidence-based interventions to deploy.

## 1. Introduction

Screening for colorectal cancer (CRC) reduces morbidity and mortality; yet, African Americans (AAs), and Latino Americans (LAs) undergo CRC screening at lower rates, relative to non-Hispanic Whites, due to accessibility and psychosocial barriers [1,2,3,4,5,6,7,8,9]. Screening disparities contribute to greater rates of late-stage diagnosis of and mortality due to CRC among these populations [6,10,11,12,13,14]. For example, in Chicago, 22.5% of LA and 35.9% of AA populations have obtained CRC screenings, relative to 38.4% among non-Hispanic Whites [15]. Mistrust in the healthcare system, for example, can result in the desire to limit interactions with healthcare facilities outside of medical necessities [16]. Mistrust can, in part, be attributed to historic and contemporary structural racism and is compounded with other ethnicity-specific factors (e.g., religiosity or customs) to influence decisions in care [17], thus complicating individual providers’ efforts to serve the underserved. This, in part, is attributed to systemic cultural and racial barriers that bar adherence to cancer screening [18,19], including CRC screening [20].

Although the American College of Physicians and the National Institutes of Health emphasize the importance of equity in healthcare access, clinician bias and inequitable access still contribute to lower-quality care [21,22,23]. Disparities in CRC screening [24] and outcomes [25] persist in LA and AA communities, respectively. To eliminate these disparities, multiple evidence-based interventions (EBIs) have been developed, including those recommended by the Guide to Community Preventive Services (Community Guide) [26], which the Center for Disease Control and Prevention (CDC) summarizes as follows: client reminders, high-quality small medial, reducing structural barriers, provider reminder and recall systems, and provider assessment and feedback [27]. Among these EBIs are culturally tailored small media (e.g., mailed birthday reminders), lay navigation (i.e., non-clinically trained personnel identifying and overcoming patient barriers to accessing care by providing support services) [28], and provider-delivered education [29,30,31,32,33,34,35]. The uptake [36,37], implementation [36], and effectiveness of select culturally tailored EBIs [36] have been examined in past research. A growing number of studies have begun to compare multiple approaches to promote CRC screening, with a focus on implementation practices, costs, and success rates [36,37,38,39].

Our study expands on this work and is among the first studies to compare the effects of multiple EBIs on screening outcomes. We are additionally among the first to consider how the impacts of EBIs across different clinical contexts, given intervention effectiveness, may vary depending on patient and provider populations [29]. Specifically, we tested the relative effectiveness of three approaches (mailed birthday reminders, lay navigation, and provider-delivered education) across clinical contexts, categorized by patient and provider ethnic compositions (multi-ethnic, predominantly AA, or predominantly LA). For our hypotheses, we leveraged the Mile Square Health Center Colorectal Cancer Awareness, Referral and Education (Mi-CARE) Program [40].

## 2. Materials and Methods

The Mi-CARE Program, a service delivery project funded by the American Cancer Society, was intended to increase CRC screening and referrals at three Chicago-area FQHCs [40]. For the purposes of this study, CRC screening included fecal immunochemical testing (FIT) and the follow-up colonoscopy after abnormal FIT result. We provide more details below, in the Measures segment of this section. A detailed study design and its rationale were previously published [40], but are described briefly below. To test the relative effectiveness of three EBIs, we conducted a prospective cross-sectional quasi-experimental study. The University of Illinois at Chicago’s Institutional Review Board and the Mile Square Health Center (MSHC) Community Advisory Board approved this study (Research Protocol #2018-0747).

Study Setting: Chicago is a city with significant CRC disparities, borne by AA and LA communities. These disparities are largely driven by historic and contemporary structural factors, including residential and resource segregation. Our study was set in three AA and LA communities that face significantly lower rates of CRC screening (47.2–61.3% vs. 71.8%), greater CRC incidence (59.1–75.12 vs. 21.8 per 100,000), and greater cancer mortality overall (190.6–210.3 vs. 149.3 per 100,000). Embedded within these vulnerable communities, MSHC represents a network of 14 FQHC clinics affiliated with the University of Illinois Hospital and Health Science System in Chicago, IL. FQHCs are U.S. Center for Medicare-certified federally funded nonprofit safety net centers, which primarily provide outpatient services to medically underserved areas or populations, regardless of their ability to pay [41] (e.g., uninsured patients). Before the implementation of the Mi-CARE Program, the CRC screening rate across the network was 13%, and provider recommendations for screening were largely opportunistic. We selected three clinic locations within the network for this study, based on the volume of patients eligible for CRC screening, their unique clinical contexts, and the hyper-localized disparities described above [40].

Study Sample: Eligible patients were each 50–75 years old, with a scheduled visit in one of the three targeted clinics during the 12-month study period. Patients were excluded if their CRC screening was up to date. Eligible patients were assigned EBIs depending on the availability of lay patient navigators and provider-delivered education during patients’ appointments [40]. For the current analysis, additional eligibility criteria included: (1) identification as AA or LA and (2) documented receipt of only one of the three interventions described below.

Staff Training: Lay patient navigators and providers completed American Cancer Society training modules regarding CRC information (e.g., risk factors, symptoms, and screening guidelines), shared decision-making, and motivational interviewing.

Design: This study was of a prospective, cross-sectional, quasi-experimental design. For this pilot study, randomization at the clinic, care, and individual levels was not possible, due to limited resources, a limited number of participating sites, FQHC system-wide practices (i.e., PCPs supporting each other), and a high potential for contamination bias.

Using these parameters, we took a quasi-experimental, opportunistic approach that leveraged variation by time and patient establishment in the FQHC. Within the first six months, established MSHC patients received mailed reminders, whereas new MSHC patients (i.e., first visit/no address available to send mailed reminders) were provided with lay patient navigation, if said patients were on site. During the last six months, PCP education was implemented for all patients. To incorporate limitations to this assignment, we removed patients who may have received multiple EPIs and incorporated the month of admission into the analyses.

The assignment of the EBIs to patients was opportunistic, based on the timing of the implementation processes and staff availability for each clinical site, as described below.

Approaches: Given our focus on intervention impacts within FQHC settings, approaches occurred within the clinical settings, or through established clinical relationships.

Mailed birthday reminders. Patients received mailed birthday reminders if they had a scheduled appointment during the study period at a participating clinic and had no opportunities for lay navigation or provider-delivered education. Lay navigators mailed an American Cancer Society postcard reminder, encouraging patients to speak to their provider or call the lay navigator to discuss CRC screening. Cards were mailed to arrive around each patient’s birthday month. Patients with mailed birthday reminders largely had their initial appointment during the first six months of study implementation, until lay navigation and provider-delivered education were available. Clinical sites differed in their abilities to implement EBIs. For example, the predominantly LA clinic was slower to implement lay navigation and provider-delivered education compared to the other two clinics. Therefore, most patients from the predominantly LA clinic received mailed birthday reminders (Table 1).

Lay navigation. Lay navigation was provided in clinical settings by racially/ethnically, linguistically, and culturally concordant non-clinical staff. FQHC-employed and -based navigators specifically provided patient education and ‘in-reach’ during clinical visits [40] and supported subsequent coordination efforts for clinical visits related to the early detection and diagnosis of CRC [28]. Lay navigation primarily occurred during the first six months of study implementation, when lay navigators were potentially available to patients. There were two navigators available across the three clinics. If lay navigators were available at the time of the appointment, patients were assigned to lay navigation. Since most patients at the predominantly LA clinic spoke Spanish, lay navigation occurred only when bicultural/bilingual navigators were available. A lay patient navigator provided education, assessed eligibility for different types of CRC testing, and engaged patients in shared decision-making concerning CRC risk and screening. If patients agreed to receive CRC screening, the patient navigator assisted with scheduling a CRC screening appointment and receipt of a medical order for a colonoscopy or FIT.

Provider-delivered education. Provider-delivered education mainly occurred during the last six months of study implementation and depended on the availability of trained providers. Fourteen providers—i.e., primary care providers (PCPs)—received training and delivered education; they primarily delivered care at the multi-ethnic and predominantly African American clinics. If lay patient navigators were unavailable at the time of the appointment, but trained providers were available, patients received provider-delivered education. Providers delivered information concerning CRC risk (e.g., via pamphlets/small media) and assessed eligibility for CRC screening. Of note, the providers in this case were primary care providers (PCPs) and did not include other clinical staff. If a patient agreed to CRC screening, the provider assisted with ordering a colonoscopy or fecal occult blood test during that same appointment.

Measures: All data were gathered in a comprehensive chart review conducted in January 2016, using the clinical network’s electronic health record (EHR) system. The primary outcome was a dichotomous (Yes/No) variable indicating completion of CRC screening within 12 months of receiving either mailed birthday reminders, provider-delivered education, or patient navigation. The independent variable was the EBI (mailed birthday reminders, lay navigation, or provider-delivered education). The reference group for all analyses was the group who received mailed birthday reminders.

The effect modifier was the clinical site, which included the multi-ethnic clinic (referent category), the predominantly LA clinic, and the predominantly AA clinic. The respective racial/ethnic compositions for the patients and providers of the selected clinics are detailed in Table 2 and Figure 1. We refer to the multi-ethnic clinic as the site where the predominant racial/ethnic group differs for patients (74% African American), and providers (60% White). We refer to the AA or LA clinics as sites where the predominant racial/ethnic groups for patients and care teams are the same, either AA (with patients and providers being 93% and 83% African American, respectively) or LA (with patients and providers both being 63% Latin American).

The demographic covariates were patient race/ethnicity, age (50–64 years old or 64–75 years old), sex, and insurance status (private insurance, public insurance, or self-payer insurance). African American race, female sex, and self-pay insurance were used as reference categories for race/ethnicity, sex, and insurance status, respectively. The month of initial appointment was included as a nominal variable, given its relationship to the types of EBIs assigned, and to account for seasonality bias.

Analysis: Analyses were conducted using SAS 9.4. Less than 1% of data were missing. In 2 cases, insurance data were unavailable, and, in 3 cases, the month of the initial appointment was unavailable. Missing information was recovered using mode-based imputation, based on the clinic for insurance status and on EBI for month of admission. We used Chi-square statistics to test differences in effectiveness among EBIs by patient and clinical factors, and to assess relationships of patient and clinical factors with the completion of CRC screening.

We examined the effects of the EBIs on CRC screening using a multivariable logistic regression model with random intercepts (GLIMMIX) to account for non-normally distributed data, correlation within patients from the same clinics, and random effects. Covariates were selected based on preliminary analyses (depicted in Table 2) and included patient race/ethnicity, age, sex, insurance status, and month of initial appointment. The month of initial appointment was included as a random effect. We assessed the clinical site as an effect modifier by evaluating the statistical significance of the coefficient associated with the product term between EBI and clinical context (EBI × clinic) in the fully adjusted model. We reported CRC screening differences stratified by clinical context and EBI. Finally, we conducted multiple sensitivity analyses, comparing in-person approaches (lay navigation versus provider-delivered education) (*n* = 851), using inverse probability of treatment weighting (IPTW) with propensity scores to address significant differences across participants assigned to different EBIs; additional analyses were conducted to examine only African American patients within the predominantly AA clinic, and only LA patients within the predominantly LA clinic.

## 3. Results

Patient demographic data are summarized in Table 1. Most patients were under 64 years old, AA, male, and publicly insured. The multi-ethnic clinic accounted for most of the patient sample. EBIs differed significantly by clinical site, patient age, patient ethnicity, and month of admission (all *p* < 0.0001).

Overall, 254 patients (20%) completed CRC screening during the study period (Table 3). Rates of screening completion significantly differed by EBI (*p* = 0.0004), clinical context (*p* < 0.0001), patient race/ethnicity (*p* < 0.0001), and the month of EBI delivery (*p* = 0.02). Patients receiving lay navigation, those in the predominantly LA clinic, patients self-identified as LA, and patients whose initial clinical appointment was within the first 4 months of the intervention period all had higher CRC screening completion rates, relative to their respective comparators (non-LA clinic, or patients, and those with appointments beyond the fourth month). Patient age, sex, and insurance status were not associated with completion of CRC screening.

In the predominantly AA clinic, lay navigation was nearly five times more effective at eliciting CRC screening completion than mailed birthday reminders (OR = 4.76, 95%CI = 1.15, 19.72, *p* = 0.03) (Table 4). In the predominantly LA clinic, provider-delivered education was one and a half times more effective at eliciting CRC screening completion relative to mailed birthday reminders (OR = 1.63, 95%CI = 1.07, 2.49, *p* = 0.02). Within the multi-ethnic clinic, provider-delivered education was half as effective at eliciting CRC screening completion relative to mailed birthday reminders (OR = 0.50, 95%CI = 0.27, 0.94, *p* = 0.03). Sensitivity analyses, using only non-imputed data and using IPTW, confirmed that lay navigation and provider-delivered education were more likely to result in CRC screening completion than mailed birthday reminders (Table 4).

Clinical context also modified the effectiveness between the two in-person EBIs (provider-delivered education or lay navigation, *n* = 851, *p* = 0.03). Lay navigation appeared more effective than provider-delivered education in the multi-ethnic and predominantly AA clinics, but the effectiveness of these methods did not differ in the predominantly LA clinic. Provider education was marginally significantly more effective than mailed birthday reminders for completion of CRC screening among LA patients in LA clinics (OR = 1.62, 95% CI = 0.97, 2.73, *p* = 0.06). There were no other significant differences between concordant of discordant pairs. Table 5 summarizes patterns in CRC screening completion by study arm across these different models.

## 4. Discussion

Cancer remains a longstanding leader among the top causes of death in the United States [42,43]. Despite an overall decline in cancer mortality, the mortality rates of preventable and treatable cancers (e.g., cervical and colorectal cancers) are higher among socioeconomically disadvantaged groups [44]. These health outcome differences have been attributed to socioeconomic disparities in access to preventive care services, such as guideline-concordant cancer screening [45]. Health disparities (i.e., differences in healthcare that adversely affect disadvantaged groups, e.g., groups with low socioeconomic status) [46] create systemic health differences along every stage of the cancer care continuum [45], including in the screening for common cancers [47], which result in delayed diagnoses and poorer outcomes for disadvantaged patient populations. The last two decades of health equity research suggest that several sources of disparity persist. For instance, the United States healthcare system is wrought with systemic racism and discrimination [48,49,50], recognized more recently by the American Medical Association as a core driver to disparities in health [51]. Social–structural factors [52] often function as determinants that systemically impact the experiences of a wide variety of groups (e.g., on the basis of age, race, or sexual orientation) [46], thus resulting in discrimination, marginalization, and, ultimately, differences in health outcomes. Furthermore, underserved communities experience disparities in health outcomes, even after accounting for insurance and income status [53], in part, due to social inequity in access to healthcare services and employment in lower-paying frontline service jobs, among other factors. Despite that decades of public health research in areas of health disparity demonstrate a critical need to move beyond the discussion of the issue of health equity and the need to address the root causes at their source [54].

The current study contributes to the existing equity-driven literature on interventions to increase CRC screening in FQHCs in several meaningful ways. Firstly, the EBIs provided in this study altogether led to higher-than-baseline rates of CRC screening (20% in the current study vs. 13% within this system prior to intervention); this finding demonstrates the need for system-level interventions to ease access to care for CRC screening and preventive health services (directly addressing the need to identify system-level interventions) [54]. This is particularly true in attesting to the utility of lay navigation, which yielded 30% adherence to CRC screening from the pre- to the post-study period. Secondly, EBIs appeared to differ in effectiveness by clinic. This finding demonstrates how EBI effectiveness differs based on the racial and ethnic demographics of both patients and providers within the clinic. More specifically, mailed birthday reminders and lay navigation EBIs appeared to be more effective in increasing the rates of CRC screening when employed in a clinic with a mixed ethnic composition. Lay navigation was somewhat more effective in increasing the rates of CRC screening in a clinic with predominantly AA patients and providers. However, provider-delivered education worked somewhat better to increase CRC screening rates within a clinic with predominantly LA patients and providers. We saw overall increases in screening in neighborhoods where screening rates were below the national average [15]. Our findings suggest that selecting optimal approaches to engage minority patients in FQHCs may differ across ethnic and racial clinical composition(s).

Several factors may aid in explaining these mixed results and extend knowledge on system interventions to improve health equity. Mailed birthday reminders appeared to be more effective than provider-delivered education in the multi-ethnic clinical site, except in sensitivity analysis using IPTW. Furthermore, our sensitivity analysis comparing the in-person EBIs also suggested that lay navigation might yield higher CRC screening rates than provider-delivered education. One potential reason for this difference is that patients’ preferences, and their ability to self-advocate in medical settings, may depend on the racial/ethnic and/or perceived identity concordance of their providers and healthcare organizations [55,56,57]. Patients may feel less confident in self-advocacy with, and have lower trust in, providers who do not share their identities, due to fear of (and previous experiences with) discrimination in these settings. Thus, they may feel more comfortable responding to an organizational letter, or to a peer who shares their identities. Conversely, if patients are engaging with providers who share their identities and/or have language concordance, they may be more confident in self-advocacy and their ability to have meaningful interpersonal contact.

Another explanation may be the overall clinical context and its implications for medical discrimination [16,57,58]. While patient–provider concordance does not always result in greater satisfaction and trust [59], an organization’s perceived commitment to hiring providers and other personnel who share identities with their patients may demonstrate that commitment to the community [60,61]. Hiring such staff may also be a proxy for the organization’s commitment to cultural competency training. Potentially, regardless of, or in addition to, the benefits of patient–provider concordance, in-person EBIs may be more effective for patients in such settings, due to greater confidence in the overall organization. Future studies are needed to explore the relationships among patient, provider, and organizational demographics, such as patient–provider concordance, to ideally situate system-level interventions to address barriers to access.

Within the AA clinic, lay navigation appeared to be more effective for CRC screening than the other two EBIs were. These findings may stem from the well-documented mistrust of medical professionals among AA patients [58,62,63]. Thus, peers may be important deliverers of EBIs for predominantly AA patients during the screening and diagnostic stages of the cancer care continuum, since these patients may be mistrustful of healthcare authorities (even racial/ethnic minority providers). Our sensitivity analyses in the predominantly AA clinic suggested no differences in effectiveness among the three interventions; however, this is likely because fewer of those patients received lay navigation. Interestingly, and of note, our provision of lay navigation services aligns with more recent advances toward a unified definition of patient navigation, which are theorized to address barriers in reimbursing navigation activities [28].

In the predominantly LA clinic, patients who received provider-delivered education were more likely to obtain CRC screening than patients who received mailed birthday reminders. This was also true for our sensitivity analyses, when using IPTW and analyzing only LA patients. This finding may reflect LA cultural values, especially their emphasis on respecting experts and adhering to their advice [64].

The current study had several limitations. Firstly, as expected, patient racial/ethnic and clinical racial/ethnic composition varied. While we adjusted for race/ethnicity and conducted analyses holding patient race/ethnicity constant, it is challenging to disentangle whether preference patterns are due to clinical composition versus patient identity. Furthermore, the study used a cross-sectional quasi-experimental design to evaluate EBIs, which may ease the burden for CRC screening access within specific minority patient populations. This precludes our paper from assessing the causal effects of the EBIs and from generalizing more broadly to the effectiveness of the selected EBIs for other racial and ethnic groups. Furthermore, our study examined several CDC-recommended EBIs in cancer screening and prevention [27]. Future research should seek to examine the relative effectiveness of all EBIs for cancer screenings for other groups known to experience health disparities (e.g., members of the LGBTQ+ community or HPV-related cancers) [65]. Regardless, this is the first-known study to extend EBIs beyond uptake and adoption and examine relative effectiveness [37], and we posit that our findings will inform future research targeting long-term implementation and translation to practice, given the current situation of racial/ethnic segregation in medical settings in the US [66,67,68,69]. Our study used a clinic-based sample of individuals receiving care within FQHCs. Thus, while AA patients disparately suffer from early-onset CRC [70], our findings may not be generalizable to populations who lack regular access, who are at risk for early CRC onset, or who do not engage with FQHCs, as different clinical resources, practices, and processes for follow-up may affect screening compliance. We could not disaggregate data by ethnicity and country of birth; thus, we could not examine variations across different subgroups of patients. In addition, our results are subject to selection bias, since patients were not randomized to the interventions. We attempted to address this issue by using the month of admission as a random effect. Future studies are warranted that can assess clinical sites’ readiness to implement such programs and incorporate models examining how clinical context may influence the effectiveness of EBIs.

The ethnic composition of patients at each clinic overlapped considerably. While this is a threat to internal validity, this overlap is common in the healthcare landscape of the U.S., due to racial/ethnic segregation in healthcare system access and utilization [68,69,70]. Furthermore, we explored only one potential effect modifier in this study (clinical site); it is likely that other factors could modify the effectiveness of EBIs. For example, a family history of CRC could influence a patient to engage in screening or to experience early-onset CRC (which is not accounted for in the current study). Finally, we could not account for other interventions that patients may have experienced, while participating in this study, through other healthcare systems or community-based organizations. We also did not ask patients about their preferences for each intervention, which could have affected their willingness to pursue CRC screening. Future work should seek to build upon the current findings to systematically test the effectiveness of EBIs in order to enhance navigation and engagement efforts through robust designs, with randomized assignment, and through the targeted implementation of system- and multi-level interventions.

## 5. Conclusions

Our evaluation of the Mi-CARE Program within a group of FQHCs suggests that the relative effectiveness of EBIs to encourage CRC screening for underserved patient populations may depend on characteristics of the EBIs, the patients, and the clinical context. Our results suggest that the implementation of interventions should be nuanced and holistic in order to be effective. Future projects should expand on these findings and deploy a more rigorous research design that: (1) uses randomization to assign EBIs; (2) compares CRC screening uptake across EBIs; and (3) incorporates patient, clinic, and provider characteristics as potential moderators or mediators of results. Future studies should test system- and multi-level interventions for health promotion to increase CRC screening in healthcare settings.

## Figures and Tables

**Figure 1 ijerph-20-07049-f001:**
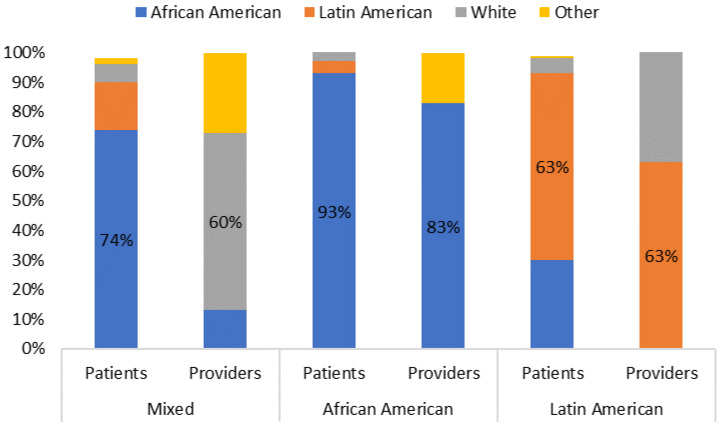
Racial and ethnic distributions of patients and providers by clinic.

**Table 1 ijerph-20-07049-t001:** Study sample characteristics by evidence-based intervention (EBI) (*n* = 1252).

	Overall (*n* = 1252)	Mailed Birthday Reminders (*n* = 401)	Lay Navigation (*n* = 86)	Provider-Delivered Education (*n* = 765)		Chi-Square
	*n* (%)	*n* (%)	*n* (%)	*n* (%)	Missing	*p*-Value
Clinical Site					0%	<0.0001
Multi-ethnic Provider	500 (40%)	80 (20%)	34 (40%)	386 (50%)		
Predominantly AA Provider	313 (25%)	50 (12%)	35 (40%)	228 (30%)		
Predominantly LA Provider	439 (35%)	271 (68%)	17 (20%)	151 (20%)		
Patient Age					0%	<0.0001
50–64	995 (79%)	295 (74%)	78 (91%)	622 (81%)		
65–75	257 (21%)	106 (26%)	8 (9%)	143 (19%)		
Patient Ethnicity					0%	<0.0001
AA	827 (66%)	184 (46%)	65 (75%)	578 (76%)		
LA	425 (34%)	217 (54%)	21 (25%)	187 (24%)		
Patient Sex					0%	0.67
Female	496 (40%)	166 (41%)	33 (38%)	297 (39%)		
Male	756 (60%)	235 (59%)	53 (62%)	468 (61%)		
Patient Insurance					<1%	0.15
Private	302 (24%)	113 (28%)	20 (23%)	169 (22%)		
Public	840 (67%)	250 (62%)	60 (70%)	530 (69%)		
Self-payer	108 (9%)	38 (9%)	6 (7%)	64 (8%)		
Month of admission					<1%	<0.0001
Months 1–4	412 (33%)	326 (81%)	75 (87%)	11 (1%)		
Months 5–8	476 (38%)	68 (17%)	11 (13%)	397 (52%)		
Months 9–12	364 (29%)	7 (2%)	0 (0%)	357 (47%)		

**Table 2 ijerph-20-07049-t002:** Racial and ethnic distributions of patients and providers by clinic.

	Mixed Clinic		African American Clinic		Latin American Clinic	
Race/Ethnicity	Patients	Providers	Patients	Providers	Patients	Providers
African American	74%	13%	93%	83%	30%	0%
Latin American	16%	0%	4%	0%	63%	63%
White	6%	60%	6%	0%	5%	38%
Other	2%	27%	2%	17%	1%	0%

**Table 3 ijerph-20-07049-t003:** Receipt of CRC screening by EBIs, effect modifiers, and covariates.

EBIs, Effect Modifiers, and Covariates	*n* (%) Received CRC Screening (*n* = 254) ^1^	Chi-Square *p*-Value
Evidence-based Intervention (EBI)		0.0004
Mailed Birthday Reminders	99 (25%)	
Provider-delivered education	129 (17%)	
Lay navigation	26 (30%)	
Clinical Site		<0.0001
Multi-ethnic	79 (16%)	
Predominantly AA	29 (9%)	
Predominantly LA	146 (33%)	
Patient Age		0.77
50–64	205 (21%)	
65–75 years old	49 (19%)	
Patient Ethnicity		<0.0001
AA	115 (14%)	
LA	139 (33%)	
Patient Sex		0.23
Female	109 (22%)	
Male	145 (19%)	
Patient Insurance		0.22
Private	70 (23%)	
Public	159 (19%)	
Self-payer	25 (23%)	
Month of admission		0.02
Months 1–4	102 (25%)	
Months 5–8	89 (19%)	
Months 9–12	63 (17%)	

^1^ The total number of individuals who obtained CRC screening was 254.

**Table 4 ijerph-20-07049-t004:** ^#^ Primary and sensitivity models on receipt of CRC screening by study arm and clinical site.

PRIMARY LOGISTIC REGRESSION MODELS (*n* = 1252)	OR	95% CI		*p*
**Evidence-based Intervention (EBI)**				
Mailed Birthday Reminders (Reference Group)	REF	REF	REF	
Provider-delivered education	1.11	0.77	1.61	0.58
Lay navigation	2.62	1.47	4.64	0.001
**Interaction Term (EBI × Clinical Context)**				0.005
Stratification by Clinical Site				
Clinic—Multi-ethnic				
Mailed Birthday Reminders (Reference Group)	REF	REF	REF	
Provider-delivered education	0.50	0.27	0.94	0.03
Lay navigation	1.22	0.47	3.15	0.68
Clinic—Predominantly AA				
Mailed Birthday Reminders (Reference Group)	REF	REF	REF	
Provider-delivered education	1.20	0.34	4.32	0.78
Lay navigation	4.76	1.15	19.72	0.03
Clinic—Predominantly LA				
Mailed Birthday Reminders (Reference Group)	REF	REF	REF	
Provider-delivered education	1.63	1.07	2.49	0.02
Lay navigation	1.90	0.70	5.17	0.21
**SENSITIVITY LOGISTIC REGRESSION MODELS #1**—**NON-IMPUTED DATA (*n* = 1247)**				
**EBI**				
Mailed Birthday Reminders (Reference Group)	REF	REF	REF	
Provider-delivered education	1.00	0.67	1.50	0.98
Lay navigation	2.01	1.20	3.64	0.009
**Interaction Term (EBI × Clinical Context)**				0.04
Stratification by Clinical Site				
Clinic—Multi-ethnic				
Mailed Birthday Reminders (Reference Group)	REF	REF	REF	
Provider-delivered education	0.50	0.27	0.94	0.03
Lay navigation	1.22	0.47	3.13	0.69
Clinic—Predominantly AA				
Mailed Birthday Reminders (Reference Group)	REF	REF	REF	
Provider-delivered education	1.20	0.34	4.32	0.78
Lay navigation	4.76	1.15	19.72	0.03
Clinic—Predominantly LA				
Mailed Birthday Reminders (Reference Group)	REF	REF	REF	
Provider-delivered education	1.63	1.07	2.50	0.02
Lay navigation	1.90	0.70	5.17	0.21
**SENSITIVITY LOGISTIC REGRESSION MODELS #2**—**USING INVERSE PROBABILITY OF TREATMENT WEIGHING (IPTW) AND PROPENSITY SCORES (*n* = 1252)**				
**EBI**				
Mailed Birthday Reminders (Reference Group)	REF	REF	REF	
Provider-delivered education	1.15	0.83	1.63	0.42
Lay navigation	3.17	1.65	6.10	0.001
**Interaction Term (EBI × Clinical Context)**				<0.0001
Stratification by Clinical Site				
Clinic—Multi-ethnic				
Mailed Birthday Reminders (Reference Group)	REF	REF	REF	
Provider-delivered education	0.76	0.35	1.63	0.48
Lay navigation	1.09	0.23	5.07	0.92
Clinic—Predominantly AA				
Mailed Birthday Reminders (Reference Group)	REF	REF	REF	
Provider-delivered education	4.31	0.58	32.00	0.15
Lay navigation	31.78	2.40	421.04	0.009
Clinic—Predominantly LA				
Mailed Birthday Reminders (Reference Group)	REF	REF	REF	
Provider-delivered education	3.49	2.30	5.29	<0.0001
Lay navigation	2.20	1.01	4.75	0.05
**SENSITIVITY LOGISTIC REGRESSION MODELS #3**—**IN-PERSON APPROACHES ONLY (*n* = 851)**				
EBI				
Provider-delivered education (Reference Group)	REF	REF	REF	
Lay navigation	2.42	1.37	4.25	0.002
**Interaction Term (EBI × Clinical Context)**				0.03
Stratification by Clinical Site				
Clinic—Multi-ethnic				
Provider-delivered education (Reference Group)	REF	REF	REF	
Lay navigation	2.40	1.04	5.41	0.04
Clinic—Predominantly AA				
Provider-delivered education (Reference Group)	REF	REF	REF	
Lay navigation	3.92	1.51	10.19	.005
Clinic—Predominantly LA				
Provider-delivered education (Reference Group)	REF	REF	REF	
Lay navigation	1.14	0.37	3.44	0.82
**SENSITIVITY LOGISTIC REGRESSION MODELS #4**—**(Patient race or ethnicity × clinical site)**				
Clinic—Predominantly AA (Only AA patients; N = 418)				
Mailed birthday reminders (Reference Group)	REF	REF	REF	
Provider-delivered education	1.52	0.53	4.34	0.44
Lay navigation	0.60	0.29	1.26	0.18
Clinic—Predominantly LA (Only LA patients; N = 336)				
Mailed birthday reminders (Reference Group)	REF	REF	REF	
Provider-delivered education	1.63	0.97	2.73	0.06
Lay navigation	1.97	0.72	5.40	0.19

*^#^ Notes.* Unless otherwise specified, we first conducted a logistic regression model with the two main effects (clinical site, approach), the one interaction term (clinical site X approach), and covariates (patient race/ethnicity, age, sex, insurance status, and month of initial appointment (included as a random effect)). If interaction terms were significant, we conducted stratified models by clinical site to examine the relative effectiveness of different approaches, with patients who received mailed birthday reminders as the reference group.

**Table 5 ijerph-20-07049-t005:** Summary of clinical site-based effect modification of the relationships between approaches and receipt of CRC screening (the symbols +, −, or = indicate that the odds of CRC screening completion by approach were either statistically significantly higher, lower, or the same as the respective referent category).

		Clinical Site		
Approach		AA	LA	Mixed
All (ref = mailed birthday reminder)	Lay	+	=	=
	Provider	=	+	−
In-person Approaches (ref = provider)	Lay	+	=	+

## Data Availability

All datasets used and/or analyzed during the current study are available from the corresponding author upon reasonable request.

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
