# Peer review of "Mi-CARE: Comparing Three Evidence-Based Interventions to Promote Colorectal Cancer Screening among Ethnic Minorities within Three Different Clinical Contexts"

_ijerph, 2023, doi:10.3390/ijerph20227049_

Round 1
Reviewer 1 Report
Comments and Suggestions for Authors
The following manuscript addresses a highly relevant topic of CRC screening among minoritized populations experiencing disparities in screening and mortality outcomes. The study examines multiple EBIs to increase screening uptake in populations at FQHCs to determine the most appropriate intervention for each group. The analyses are very thorough to determine the best intervention while recognizing the complexity and context of the study environment.
Addressing the following suggestions would improve the manuscript prior to publication:
1. The introduction could be condensed, it needs more emphasis on existing evidence of these EBIs for CRC screening. The first two paragraphs could better fit in the discussion rather than the introduction.
2. Are their disparities for CRC in these two populations, for example? Do they screen at lower rates than national averages? Do these populations experience excess mortality due to CRC? Is there context specific epi data that describes the impact on these groups in Chicago? The authors might consider including these data in the first paragraph to set the stage for the study.
3. Have there been similar studies with multiple approaches to examine CRC screening uptake? Or perhaps similar approaches for other cancer sites?
4. What type of screening are we referring to? FIT tests, colonoscopy, etc?
5. Why use lay navigators? The varying definitions of patient navigators v. community health workers, etc. has led to some to call for more stringent definitions and greater professionalization of these roles:
https://acsjournals.onlinelibrary.wiley.com/doi/full/10.3322/caac.21805
6. Who exactly were the trained providers? Nurses? Other clinical staff?
7. Did the authors consider randomizing by sites and by type of intervention? Could a patient receive one or more, for example, a birthday reminder and patient education? If the patient screened, how do you account for the EBI that motivated uptake?
8. What about early onset CRC? While the study highlights guideline specific screening, the discussion could be improved by recognizing the dramatic increase in CRC incidence in younger individuals not aligned with current guidelines.
Author Response
Reviewer #1
The following manuscript addresses a highly relevant topic of CRC screening among minoritized populations experiencing disparities in screening and mortality outcomes. The study examines multiple EBIs to increase screening uptake in populations at FQHCs to determine the most appropriate intervention for each group. The analyses are very thorough to determine the best intervention while recognizing the complexity and context of the study environment.
Thank you for the vote of confidence in our study as well as the thoughtful critique of our manuscript. We address each of the comments below:
Addressing the following suggestions would improve the manuscript prior to publication:
1. The introduction could be condensed, it needs more emphasis on existing evidence of these EBIs for CRC screening. The first two paragraphs could better fit in the discussion rather than the introduction.
We thank the reviewer for this suggestion. We have included more evidence on existing EBIs (see p. 2 lines 47-54 and have moved the first two paragraphs into the discussion section (see p. 9, lines 264-286).
2. Are there disparities for CRC in these two populations, for example? Do they screen at lower rates than national averages? Do these populations experience excess mortality due to CRC? Is there context specific epi data that describes the impact on these groups in Chicago? The authors might consider including these data in the first paragraph to set the stage for the study.
Thank you for the constructive feedback. For the introduction, we provide additional references and succinctly describe population disparities in CRC screening in Chicago. In the Introduction, Method, and Discussion sections, we describe context-specific epi data, with a focus on the specific Chicagoland Latino and African American communities that were served as part of this study.
Introduction, p. 2 lines 47-54, Method, p. 3 lines 107-109, and Discussion, p.10, lines 321.
Have there been similar studies with multiple approaches to examine CRC screening uptake? Or perhaps similar approaches for other cancer sites?
Thank you for this feedback. We now incorporate several more studies in the Introduction and Discussion section, which address multiple approaches to CRC screening. While we did not find studies specific to relative effectiveness of EBIs, we’ve now referenced several studies on implementation and effectiveness in Introduction (p. 2, lines 71-75). We further welcome any additional references and citations that the reviewer may recommend. To summarize succinctly, from what we are aware, comparative research on EBIs has largely focused on implementation practices (Adams et al., 2018), cost (Kim et al., 2021), extent of implementation of EBIs (Sharma et al., 2021). We believe that we are among the first studies to compare the effects of EBIs on screening outcomes and to evaluate the impact of EBIs by clinical context. We would however be grateful to incorporate any additional relevant literature that the reviewer recommends.
- What type of screening are we referring to? FIT tests, colonoscopy, etc?
Our CRC screening refers to the use of FIT tests for screening and colonoscopies for the follow up on any abnormal FIT tests. We have attempted to clarify this in the method section on p. 2, lines 87-89.
- Why use lay navigators? The varying definitions of patient navigators v. community health workers, etc. has led to some to call for more stringent definitions and greater professionalization of these roles:
https://acsjournals.onlinelibrary.wiley.com/doi/full/10.3322/caac.21805
Thank you for sharing this paper and for this comment. We recognize the ambiguity of patient navigation definitions in past research and have revised the paper for greater clarity to contribute more meaningfully to future research. This includes use of an updated definition of lay navigation aligned with this seminal Chan et al. 2023 review (Introduction, p. 2, lines 69-71), clarification of our approach (Method, p. 3, lines 148-152), and potential contributions (Discussion, p. 10, lines 339-342). For our study, we now justify the use of lay navigators with our scope, which was to compare the effectiveness of interventions based in clinical settings and through clinical relationships. We refer to the previously published protocol referenced in our study’s method section (reference 41), which describes in detail how lay navigators, who were non-clinical staff, were employed by the FQHC and integrated into FQHCs.
- Who exactly were the trained providers? Nurses? Other clinical staff?
PCPs were the subjects for training. As such, nurses and other clinic staff were not included in the mention of “trained providers”. We have attempted to clarify this in our paper on page 4, lines 164-166.
7. Did the authors consider randomizing by sites and by type of intervention? Could a patient receive one or more, for example, a birthday reminder and patient education? If the patient screened, how do you account for the EBI that motivated uptake?
We thank the reviewer for this question. Randomization was considered, but ultimately not feasible due to the scope of this pilot project, with a limited number of participating sites and high potential for contamination bias during implementation. There were a few patients who received multiple EBIs (mailed letter + navigation; mailed letter + provider education), who were removed from analyses. We have attempted to clarify this within the revised manuscript’s Method (p. 3, lines 111-116).
What about early onset CRC? While the study highlights guideline specific screening, the discussion could be improved by recognizing the dramatic increase in CRC incidence in younger individuals not aligned with current guidelines.
For patients to be eligible to receive an EBI, they had to be of age for regular CRC screening. However, we have noted the need for future work in early detection as part of our Discussion section (see p. 10, lines 364-367).
Reviewer 2 Report
Comments and Suggestions for Authors
Thank you for the opportunity to review this paper, describe results of a comparison of three evidence-based interventions to improve colorectal cancer screening. Overall, the manuscript is well written, clear, and makes an important contribution to the cancer disparities research literature. I have a few comments and suggestions for improvement.
Abstract: The study setting (US) was not immediately obvious in the abstract. Perhaps clarify this.
Background:
-Line 75: "complicating the role of diversity in serving the underserved". It is not clear what this means.
-Line 79: "subpar access" - consider describing as disparate or inequitable access to convey the true nature of the differences in access to healthcare
-Lines 81-82: references three evidence-based interventions. Are there others that are relevant beyond the three that were tested in this study? Could be helpful to identify the range of options, even if they were not directly tested
Methods:
-The study design was not explicitly stated
-Some additional context would be helpful in terms of the setting. What is a FQHC? Who most often accesses care here? Perhaps this could be briefly added to the study setting (and would be especially helpful for readers outside the US who may struggle to understand the complex system of public and privately funded healthcare, and who qualifies for public insurance).
-The assignment of participants to the intervention(s) was unclear. Was there a systematic process? Was assignment opportunistic? A diagram or algorithm might be helpful to add clarity.'
Discussion:
-Generally a robust and interesting discussion of findings. Much of the discussion focused on ethnicity as a main factor, but I wonder about the role of other 'patient identity' factors beyond ethnicity (e.g., economic status, housing status, etc)? Perhaps these could be addressed in the discussion as well.
Author Response
Reviewer #2
Abstract: The study setting (US) was not immediately obvious in the abstract. Perhaps clarify this.
We have provided this clarification.
Background:
-Line 75: "complicating the role of diversity in serving the underserved". It is not clear what this means.
Thank you. We have rephrased this sentence.
-Line 79: "subpar access" - consider describing as disparate or inequitable access to convey the true nature of the differences in access to healthcare.
Thanks for the suggestion. We have reworded this to ‘inequitable access’.
-Lines 81-82: references three evidence-based interventions. Are there others that are relevant beyond the three that were tested in this study? Could be helpful to identify the range of options, even if they were not directly tested
Excellent suggestion. We have now included other EBIs endorsed by the Centers for Disease Control and Prevention, stemming from the Community Preventive Services Taskforce, in our Introduction (see p. 2, lines 63-68).
Methods:
-The study design was not explicitly stated
We have described the study as a “cross sectional prospective quasi experimental” design in materials and methods (p. 2, lines 91-92) and also now include a section in the Methods for discussing our design choices.
-Some additional context would be helpful in terms of the setting. What is a FQHC? Who most often accesses care here? Perhaps this could be briefly added to the study setting (and would be especially helpful for readers outside the US who may struggle to understand the complex system of public and privately funded healthcare, and who qualifies for public insurance).
Great suggestion; we have now added this definition to the method section(see p. 2, lines 102-105).
-The assignment of participants to the intervention(s) was unclear. Was there a systematic process? Was assignment opportunistic? A diagram or algorithm might be helpful to add clarity.'
Thank you. We have now explicitly labeled patient assignment to EBIs as opportunistic (see p. 3, 124-132) and noted this as a limitation of our study.
Discussion:
-Generally a robust and interesting discussion of findings. Much of the discussion focused on ethnicity as a main factor, but I wonder about the role of other 'patient identity' factors beyond ethnicity (e.g., economic status, housing status, etc)? Perhaps these could be addressed in the discussion as well.
Thanks! We have added some other factors related to patient identity (e.g., identification as a member of the LGBTQ+ community) that could make a difference in having full and comprehensive whole person patient needs addressed (see p.10, 358-359).